# Photo-induced ring-maintaining hydrosilylation of unactivated alkenes with hydrosilacyclobutanes

Shaowei Chen, Meiyun Gao, Xiaoqian He & Xiao Shen ◉ ✉

Increasing attention has been paid to silacyclobutanes because of their wide application in ring opening and ring extension reactions. However, the synthesis of functionalized silacyclobutanes remains an unmet challenge because of the limited functional group tolerance of the reactions with organometallic reagents and chlorosilacyclobutanes. Herein, we report a conceptually different solution to this end through a visible-light-induced metal-free hydrosilylation of unactivated alkenes with hydrosilacyclobutanes. A wide range of unactivated alkenes with diverse functional groups including the base-sensitive acid, alcohol and ketones participated in this reaction smoothly. In particular, the first hydrosilylation reaction of alkenes with dihydrosilacyclobutane provides a facile access to various functionalized alkyl monohydrosilacyclobutanes. Unsymmetrical dialkyl silacyclobutanes have also been synthesized through consecutive hydrosilylation with dihydrosilacyclobutane in one pot. The mechanism study reveals that the Lewis basic solvent could promote the generation of strained silyl radicals by direct light irradiation without a redox-active photocatalyst and the thiol catalyst plays an important role in accelerating the reaction.

Organosilicon compounds are of great value in organic synthesis, medicinal chemistry and material science[1–3]. As a kind of important organosilicon reagents, silacyclobutanes (SCBs) have attracted much attention in recent years[4–33]. Taking advantage of the high ring tension (strain energy: 24.5 kcal/mol)[8], diverse ring opening and ring extension reactions of SCBs have been developed to synthesize a wide range of important organosilicon compounds (Fig. 1a)[4–32]. Although SCBs have emerged as lynchpins in the synthesis of organosilicon compounds, the synthesis of functionalized SCBs is challenging. Traditional synthesis of 1,1-disubstituted SCBs relied on the nucleophilic substitution reaction of 1-chlorine-substituted silacyclobutanes with organometallic reagents (Fig. 1b)[4–11]. Most of the substituents introduced by this method are simple aryl groups and non-functionalized alkyl groups, such as ethyl and *n*-butyl, because of the limited functional group tolerance of the basic and nucleophilic organometallic reagents. Recently, Zhao and coworkers reported a seminal method for the

synthesis of SCBs through a Ni-catalyzed reductive coupling reaction between 1-chloro-substituted silacyclobutanes and aryl/vinyl halides/pseudo-halides, which bypassed the pre-synthesis of organometallic reagents (Fig. 1b)[33]. However, the corresponding reactions with alkyl halides/pseudo-halides were not successful, and the 1-chloro-substituted silacyclobutanes used in these reactions still need to be synthesized from organometallic reagents. To the best of our knowledge, there is no general method for the synthesis of alkyl SCBs with broad functional group diversities. As a step- and atom-economical approach to synthesizing organosilicon compounds, hydrosilylation of alkenes has been extensively studied[34–46]. The most widely studied olefin hydrosilylation reactions are based on transition-metal catalysis[1,34,35], but recent visible-light-induced photocatalytic reactions through the generation of silyl radicals provided a conceptually different approach for organosilicon compounds synthesis (Fig. 1c)[36–46]. In particular, the employment of organophotocatalyst to

The Institute for Advanced Studies, Engineering Research Center of Organosilicon Compounds & Materials, Ministry of Education, Wuhan University, 299 Bayi Road, 430072 Wuhan, Hubei, PR China. ✉e-mail: xiaoshen@whu.edu.cn

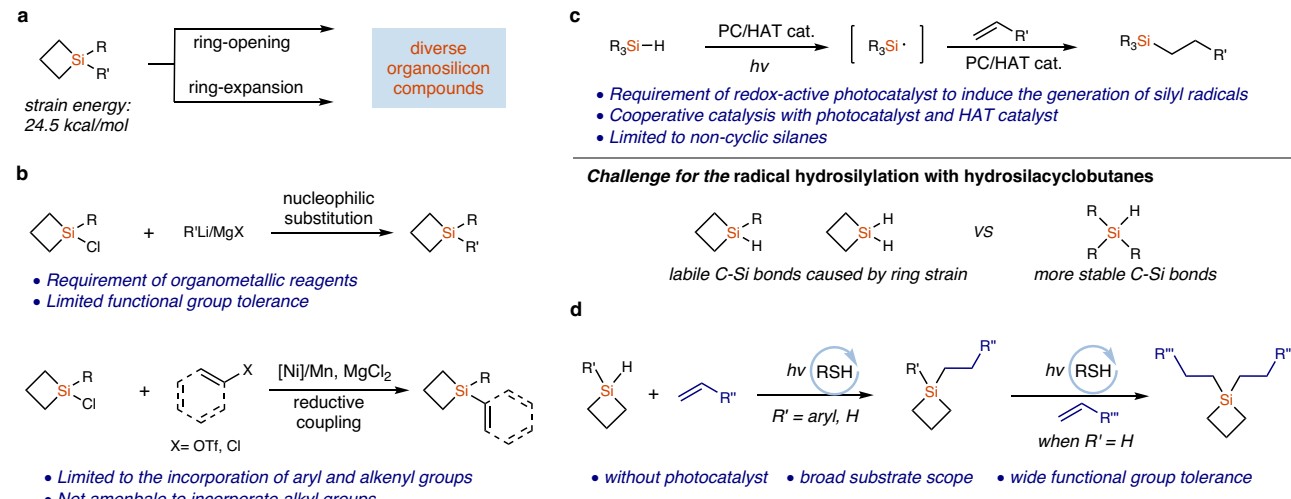

**Fig. 1 | Background and our strategy for the synthesis of functionalized silacyclobutanes. a** Silacyclobutanes are important lynchpin intermediates for the synthesis of value-added organosilicon compounds through ring-opening and ring-expansion reactions. **b** Previous methods for the synthesis of silacyclobutanes relied on the nucleophilic substitution reactions with organometallic reactions and transition-metal catalyzed reductive coupling reactions, but they are not applicable in the synthesis of functionalized alkyl SCBs. **c** Previous photocatalytic radical hydrosilylation of alkenes was limited to linear hydrosilanes, and the radical hydrosilylation with hydrosilacyclobutanes is more challenging because of the labile C–Si bonds of the strained four-membered ring compounds. **d** This study: photo-induced metal-free radical hydrosilylation of alkenes with broad substrate scope and excellent functional group tolerance, in the absence of any redox-active photocatalyst.

induce the generation of silyl radicals bypassed the intermediacy of transition-metals[37–42]. Despite these advances, photocatalytic hydrosilylation reactions are still limited to non-cyclic hydrosilanes (Fig. 1c). The radical hydrosilylation with HSCBs is more challenging because the strained rings could lead to problematic C–Si bond cleavage side reactions.

In this work, we report a unique photo-induced metal-free radical hydrosilylation of unactivated alkenes without redox active photocatalyst, which avoids the photocatalyst-induced decomposition of SCBs (Fig. 1d). The practical method significantly broadens the diversity of SCBs, enhancing the research of using SCBs as lynchpins in the synthesis of organosilicon compounds. In particular, the first successful application of dihydrosilacyclobutane (DHSCB) in hydrosilylation reactions not only provides a general synthesis of functionalized alkyl monohydrosilacyclobutanes (MHSCBs) but also provides a facile synthesis of unsymmetrical dialkyl SCBs through sequential hydrosilylation in one pot.

## Results

### Reaction development

Since the ketone-containing 1,1-disubstituted SCB **3a** cannot be prepared by previous methods employing organometallic reagents or through the transition-metal-catalyzed reactions, MHSCB **1a** and alkene **2a** were chosen as model substrates to test the idea of hydrosilylation of alkenes with MHSCBs to prepare SCBs (Table 1). Inspired by Wu's seminal work[37], we first tested the employment of 5 mol% of eosin Y as the photocatalyst in the hydrosilylation reaction between **1a** (1.0 equiv.) and **2a** (1.0 equiv.), in the presence of 10 mol% of *i*-Pr₃SiSH as the co-catalyst, under 460 nm in 1,4-dioxane, but only 36% yield of **3a** was observed by ¹H NMR, and significant decomposition of MHSCB **1a** was observed (Table 1, entry 1). Replacement of neutral eosin Y with eosin Y-Na as the photocatalyst did not improve the reaction (100% conversion of **1a**, 38% yield of **3a**, Table 1, entry 2). Fortunately, when the catalyst was changed to TBADT or benzophenone and the light was changed to 390 from 460 nm, the yield of **3a** was significantly improved (64%, Table 1, entry 3; 85%, Table 1, entry 4). However, the control experiment showed that 85% yield of **3a** could also be obtained under 390 nm without a photocatalyst, albeit the corresponding reactions at 420 and 460 nm led to significantly decreased yield

(Table 1, entries 5–7). The reaction did not proceed without light irradiation (Table 1, entry 8). Further investigation of the reaction conditions without adding a photocatalyst by changing the solvent revealed that THF was the best solvent (Table 1, entries 9–13). The less sterically hindered THF is better than the more sterically hindered Et₂O and MeOᵗBu (Table 1, entries 9–11). The use of less coordinating PhMe as the solvent led to only a 2% conversion of **1a**. It is known that the Lewis acidity of SCBs (Denmark's strain release Lewis acidity) is higher than that of linear silanes[27,47–50]. The above results indicate that the coordination of solvent to the Lewis acidic SCBs could promote the hydrosilylation reaction. However, the reaction in CH₂Cl₂ resulted in a 54% conversion of **1a**, but only a 2% yield of **3a** was obtained because of the competing chlorine atom abstraction reaction by the silyl radical from CH₂Cl₂ (Table 1, entry 13). The chloro derivative of **1a** was observed by Liu et al.[29] Si NMR and quenched by MeMgBr, affording 1-methyl-1-(*p*-tolyl)siletane **1a-1** in 63% yield (see Supplementary Fig. 3). With THF as the solvent, slightly increasing the amount of **1a** to 1.2 equivalents and extending the reaction time to 36 h result in the optimal conditions for the preparation of **3a** (95% yield, Table 1, entry 14).

### Substrate scope exploration

With the optimized conditions, we examined the scope of unactivated alkenes **2**. As shown in Fig. 2, mono-substituted alkenes bearing various functional groups, including ketone, free alcohol, carboxylic acid, carboxylic ester, aldehyde, nitrile, chloride, SiMe₃, B(Pin), ether and epoxide, were all well tolerated, giving the corresponding SCBs **3a–3m** in 61–96% yields. Selective hydrosilylation of the terminal alkene in the presence of an internal alkene was achieved without altering the reaction conditions, and compound **3n** was isolated in 57% yield. If a diene containing two terminal alkenes was employed, decreasing the amount of MHSCB **1a** to 1.0 equivalent was needed to suppress the bis-silylation side reaction, and compound **3o** was isolated in 64% yield. When the amount of **1a** was increased to 2.4 equivalents, a successful bis-silylation was achieved, affording compound **3p** in 74% yield. Moreover, electron-rich vinyl ethers, vinyl ester, and vinyl sulfide could also participate in the reactions to produce products **3q–3t** in 61–95% yields. Vinyl amides and vinyl carbazole also worked well in the reactions with MHSCB **1a**, affording the corresponding hydrosilylation

**Table 1 | Screening the reaction conditions for the hydrosilylation of unactivated alkenes with hydrosilacyclobutanes[a]**

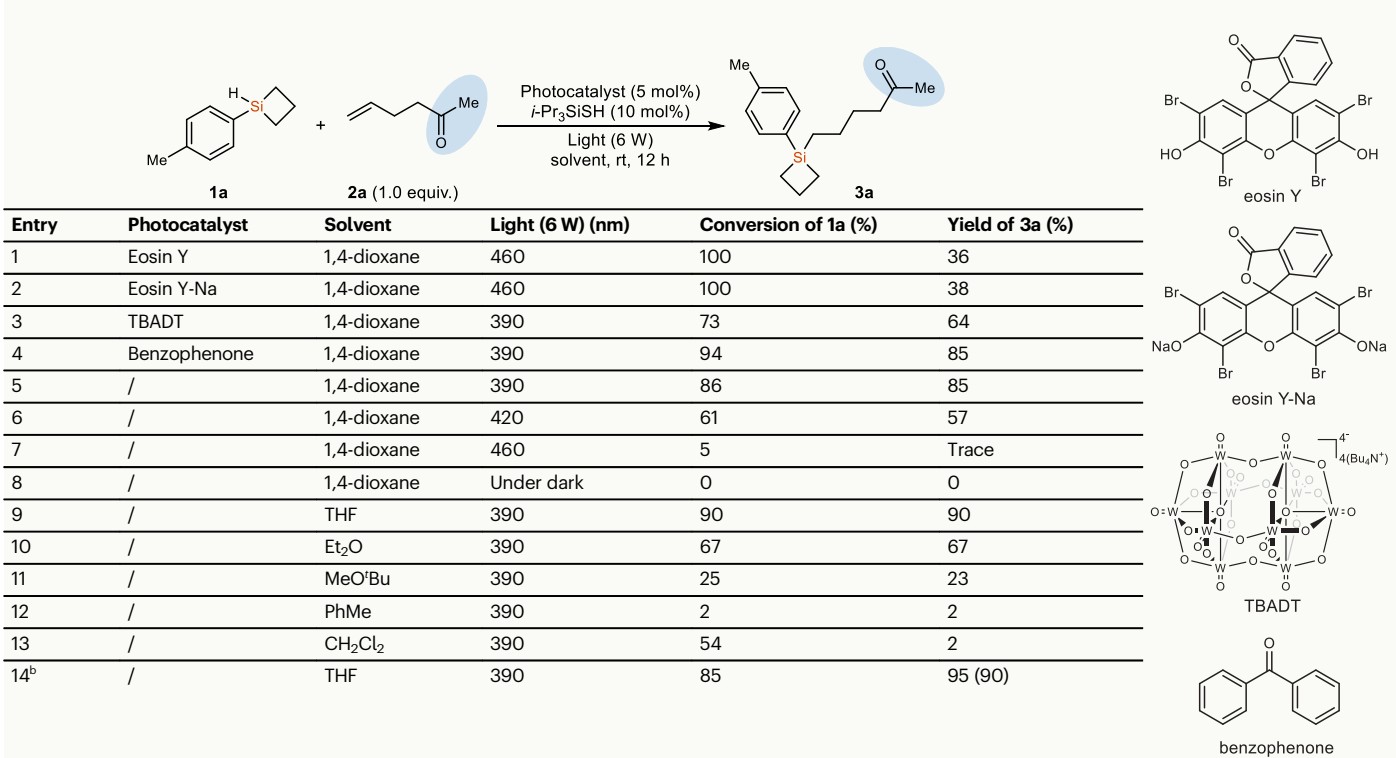

| Entry | Photocatalyst | Solvent | Light (6 W) (nm) | Conversion of 1a (%) | Yield of 3a (%) |
|---|---|---|---|---|---|
| 1 | Eosin Y | 1,4-dioxane | 460 | 100 | 36 |
| 2 | Eosin Y-Na | 1,4-dioxane | 460 | 100 | 38 |
| 3 | TBADT | 1,4-dioxane | 390 | 73 | 64 |
| 4 | Benzophenone | 1,4-dioxane | 390 | 94 | 85 |
| 5 | / | 1,4-dioxane | 390 | 86 | 85 |
| 6 | / | 1,4-dioxane | 420 | 61 | 57 |
| 7 | / | 1,4-dioxane | 460 | 5 | Trace |
| 8 | / | 1,4-dioxane | Under dark | 0 | 0 |
| 9 | / | THF | 390 | 90 | 90 |
| 10 | / | Et$_2$O | 390 | 67 | 67 |
| 11 | / | MeO$^t$Bu | 390 | 25 | 23 |
| 12 | / | PhMe | 390 | 2 | 2 |
| 13 | / | CH$_2$Cl$_2$ | 390 | 54 | 2 |
| 14[b] | / | THF | 390 | 85 | 95 (90) |

THF tetrahydrofuran.

[a]Reaction conditions: **1a** (0.1 mmol), **2a** (0.1 mmol), photocatalyst (5 mol%), $i$-Pr$_3$SiSH (10 mol%), light (6 W), solvent (1 mL), 12 h.

[b]**1a** (0.6 mmol), **2a** (0.5 mmol), $i$-Pr$_3$SiSH (10 mol%), 390 nm (6 W), THF (2 mL), 36 h; isolated yield is given in the parenthesis.

products **3u**–**3x** in 77–92% yields. In addition, aryl hydrosilacyclobutanes possessing different substituents (F, $t$-Bu, NMe$_2$) on the aromatic rings also participated in the reaction smoothly to generate **3y**–**3ab** in 77–93% yields. It is worth noting that the 1,1-disubstituted alkene was well accommodated and compound **3ac** was isolated in 77% yield with excellent anti-Markovnikov regioselectivity. Mono-substituted alkenes derived from menthol and DHEA could also undergo the hydrosilylation smoothly to afford products **3ad** and **3ae** in 74% yield and 75% yield, respectively.

After achieving the efficient synthesis of 1,1-disubstituted SCBs with MHSCBs, we subsequently studied the selective synthesis of alkyl MHSCBs through radical hydrosilylation reaction with DHSCB (**1f**), because alkyl MHSCBs that contain functional groups are also difficult to synthesize by previous methods (Fig. 3). Compound **1f** is a gas that has not been explored in synthetic chemistry, probably because of its difficulty to access[51–55]. We first developed an efficient synthesis of **1f** with 77% yield that can be stored as a stock solution in Et$_2$O through the reduction of commercially available 1,1-dichlorosiletane with LiAlH$_4$ (for details, see Supporting Information). With compound **1f** as the reaction partner, aliphatic alkenes bearing various functionalized groups, including Ph, CO$_2$H, Cl, OH, OAc, OCOPh, and Bpin, worked well, giving the corresponding MHSCBs **3af**–**3al** in 72–91% yields. Vinyl carbazole was also well tolerated in this reaction, giving the product **3am** in 75% yield. The alkene derived from L-menthol also reacted well with dihydrosilacyclobutane **1f**, affording product **3an** an 86% yield. Diene **2g**, which contains two terminal double bonds participated in this reaction smoothly, and bissilylation product **3ao** was isolated in 77% yield. The reactions with internal alkene such as ($Z$)-cyclooctene and 1,1-disubstituted alkene such as (3-methylbut-3-en-1-yl)benzene worked well, giving corresponding hydrosilylation products **3ap** and **3aq** in 50% and 67% yield, respectively.

Encouraged by the above results, we then studied the synthesis of unsymmetrical dialkyl SCBs through sequential hydrosilylation of alkenes in one pot (Fig. 3). Reagent **1f** is volatile, and the excess amount of **1f** can be easily removed from the reaction mixture, thus no column purification of the first hydrosilylation product is needed, and the reaction mixture can be directly used in the next hydrosilylation process after removal of **1f** under vacuum. Under the simple procedures, various unsymmetrical dialkyl SCBs have been prepared. Alkenes bearing various functional groups, including carboxylic ester, Cl, OH, Bpin, CN, and carbazole were all well tolerated, giving the corresponding products **4aa**–**4ag** in 68–86% yields. Moreover, the one-pot consecutive hydrosilylation reaction could also be applied in the late-stage functionalization of complex alkenes, affording the derivatives from L-menthol, Citronellol, Cholesterol, and DHEA (Dehydroepiandrosterone) in synthetically useful yields (**4ah**–**4ak**, 43–75% yields).

## Synthetic applications

To show the synthetic potential of this method, we scaled up the reaction to 5 mmol scale, and SCB **3s** was isolated in 83% yield (1.087 g, Fig. 4a). Thanks to the ring tension, silacyclobutanes could undergo diverse ring opening or ring expansion reactions to access a wide variety of functionalized organosilicon compounds. For example, **3s** could undergo a ring-opening reaction with styrene to give $E$-alkenylsilane **5** in 86% yield through the Ni-catalyzed reaction[12]. In the presence of a Pd-catalyst, ethyl buta-2,3-dienoate underwent [4 + 2] cycloaddition with SCB **3s**, affording 2-($E$)-enoate-substituted silacyclohexene **6** in 85% yield[23]. The Rh-catalyzed intermolecular C($sp$)–H bond silation reaction of a terminal alkyne with compound **3s** also performed well, giving alkynylsilane **7** in 80% yield[56]. Treatment of **3s** with diphenyl acetylene under Ni-catalyzed conditions afforded silacene **8** in 73% yield[57]. Moreover, the reaction of

compound **3af** with MeOH in the presence of an *N*-heterocyclic carbene catalyst afforded siloxane **9** in 73% yield[58]. In addition, the hydrosilylation of activated alkenes with MHSCBs was also achieved,

affording unsymmetrical dialkyl silacyclobutanes **10a–10c** in 55–81% yields, in the presence of 4CzIPN (2,4,5,6-Tetra(9*H*-carbazol-9-yl) isophthalonitrile) as the photocatalyst.

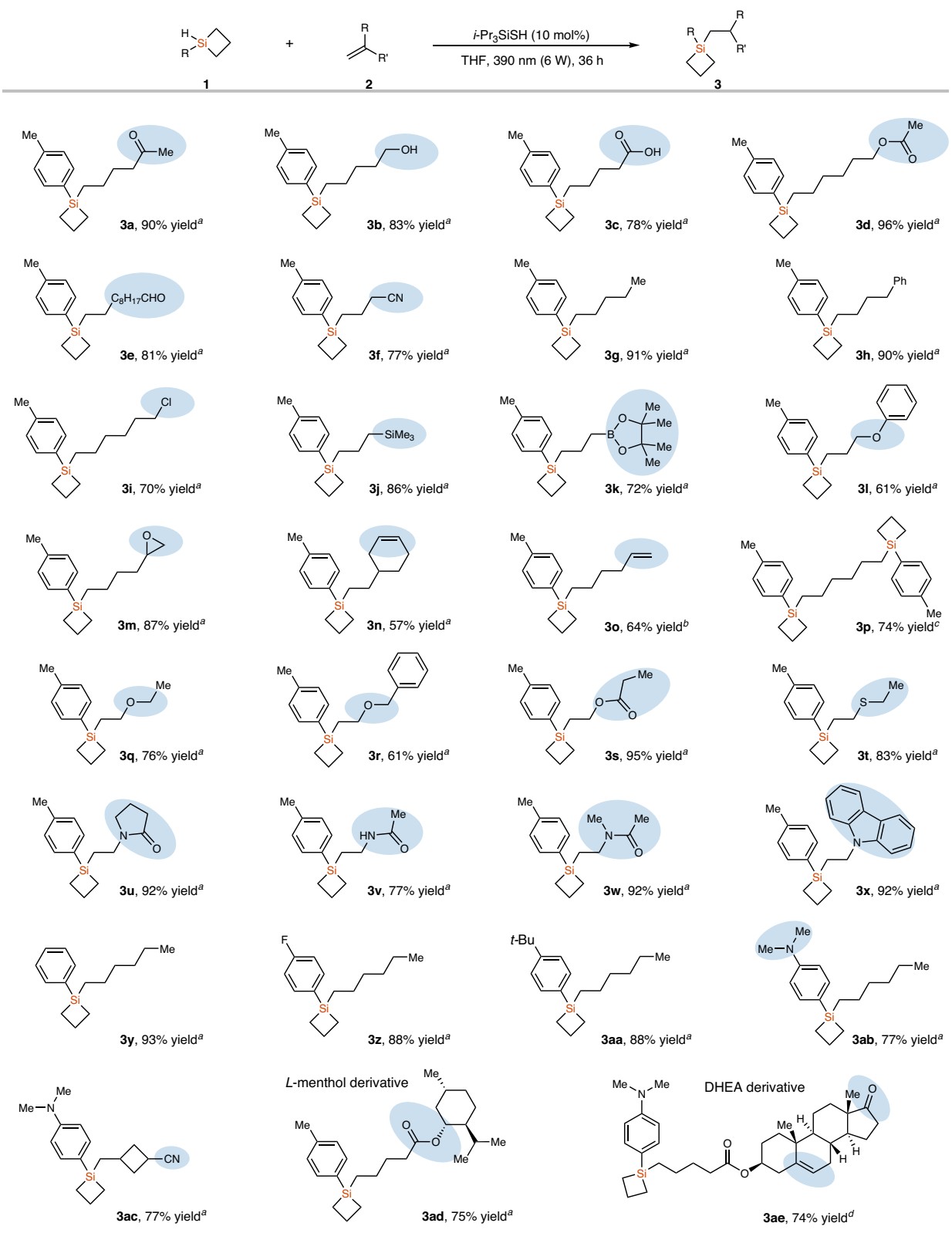

**Fig. 2 | Substrate scope for the hydrosilylation of alkenes with MHSCBs.** [a]**1** (0.6 mmol), **2** (0.5 mmol), *i*-Pr₃SiSH (10 mol%), THF (2 mL), 36 h. [b]**1a** (0.5 mmol), **2o** (0.6 mmol), *i*-Pr₃SiSH (10 mol%), THF (2 mL), 36 h. [c]**1a** (1.2 mmol), **2o** (0.5 mmol), *i*-

Pr₃SiSH (10 mol%), THF (2 mL), 60 h. [d]**1a** (0.75 mmol), **2aa** (0.5 mmol), *i*-Pr₃SiSH (10 mol%), THF (2 mL), 48 h.

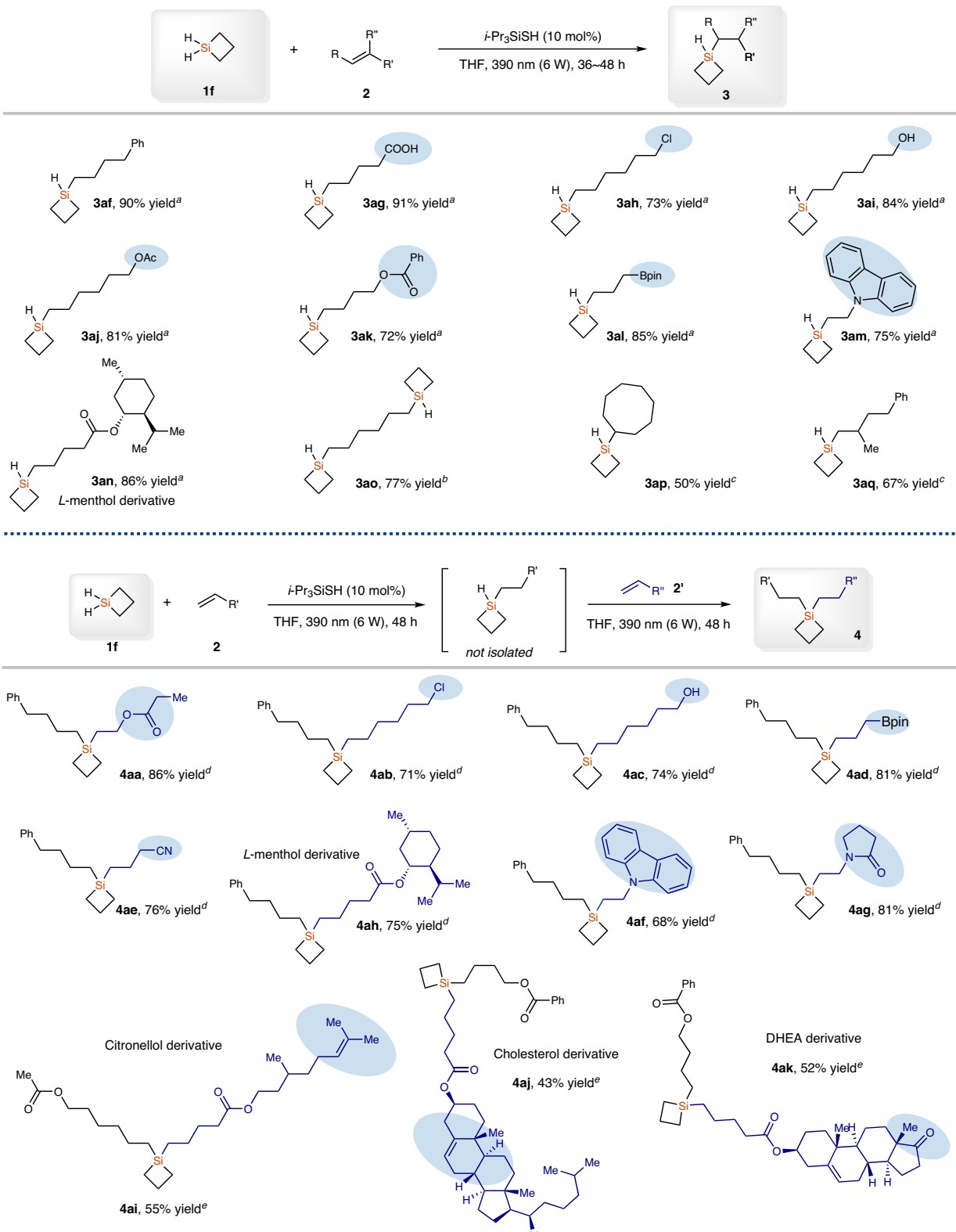

**Fig. 3 | Substrate scope for the hydrosilylation of alkenes with DHSCB (1f).** [a]**1f** (0.76 mmol, 0.30–0.4 M in Et₂O), **2** (0.5 mmol), *i*-Pr₃SiSH (10 mol%), THF (1 mL), 390 nm (6 W), 36–48 h. [b]**1f** (1.52 mmol, 0.30–0.4 M in Et₂O), **2o** (0.5 mmol), *i*-Pr₃SiSH (10 mol%), THF (1 mL), 390 nm (6 W), 60 h. [c]**1f** (1.14 mmol, 0.30–0.4 M in Et₂O), **2** (0.5 mmol), *i*-Pr₃SiSH (10 mol%), THF (1 mL), 390 nm (6 W), 48 h. [d]**1f** (1.14 mmol, 0.30–0.4 M in Et₂O), 2 (0.75 mmol), *i*-Pr₃SiSH (10 mol%), THF (1 mL),

390 nm (6 W), 48 h; then the mixture was concentrated under vacuum and **2′** (0.5 mmol) and THF (2 mL) was added and the resulting mixture was stirred under 390 nm (6 W) for 48 h. [e]**1f** (1.52 mmol, 0.30–0.4 M in Et₂O), **2** (1 mmol), *i*-Pr₃SiSH (10 mol%), THF (1 mL), 390 nm (6 W), 48 h; then the mixture was concentrated under vacuo and **2′** (0.5 mmol), *i*-Pr₃SiSH (5 mol%) and THF (2 mL) was added and the resulting mixture was stirred under 390 nm (6 W) for 48 h.

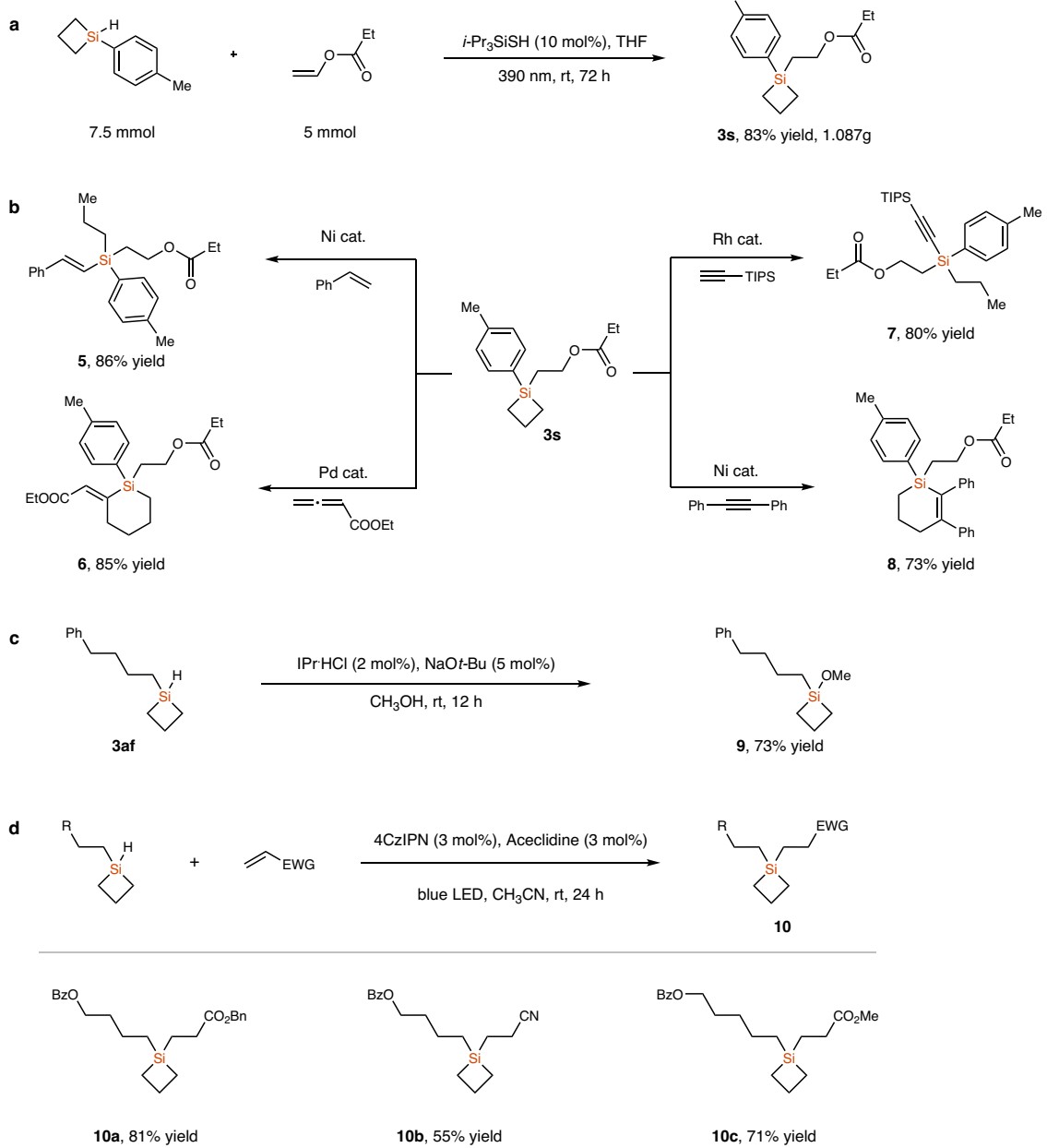

**Fig. 4 | Gram scale synthesis and synthetic applications. a** Gram scale synthesis of SCB **3s**. **b** Ring-opening and ring-expansion reactions of SCB **3s**; "cat.": catalyst. **c** Synthesis of silyl ether **9**. **d** Photocatalytic hydrosilylation of activated alkenes with a monohydrosilacyclobutane.

## Mechanistic studies

In order to shine some light on the mechanism of this reaction, several control experiments have been conducted. Under standard conditions, non-strained silacyclopentane (**1g**) and acyclic Et₂SiH₂ have been tried, both the conversion of **2h** and the yield of product decreased compared with the reaction of **1f**. Moreover, the reaction with Et₃SiH was much slower, affording compound **13** in only 5% yield. These results indicate that the existence of ring leads to higher reactivity of HSCB (Fig. 5a) Under the standard conditions, we found that the product **3h** could be detected without the addition of thiol catalyst, albeit the reaction was slow and only 11% yield was obtained after 120h, indicating that silyl radical could be generated by direct light irradiation of **1a** (Fig. 5b). Alkene **2ah** was then used to conduct a radical clock experiment. When the reaction was performed under the standard conditions, compound **14** was isolated in 41% yield, and compound **15** was isolated in 81% yield (Fig. 5c), supporting the generation of a silyl

radical and a thiyl radical in the reaction. Without *i*-PrSiSH, compound **14** was generated in 5% yield. The reaction of *i*-PrSiSH and **2ah**, in the absence of **1a**, afforded compound **15** an 85% yield. Furthermore, EPR experiments were performed to investigate the generation of silyl radical intermediate by light irradiation of **1a**. With the addition of radical spin trapping reagent DMPO (5,5-dimethyl-1-pyrroline-*N*-oxide), a mixture signal of possible silyl radicals ($g = 2.0076$, $A_N = 14.30$ G, $A_H = 20.60$ G), carbon radicals ($g = 2.0076$, $A_N = 13.90$ G, $A_H = 18.0$ G) and hydrogenated DMPO ($g = 2.0076$, $A_N = 14.50$ G, $A_{H1} = 18.80$ G, $A_{H2} = 18.80$ G) were identified (Fig. 5d)[59]. EPR studies of *i*-Pr₃SiSH and the reaction system were also conducted (for details, see Supplementary Fig. 4). These results provided strong evidence for the radical process of this reaction. To investigate the possible coordination between THF and HSCB, we conducted the NMR experiments, and the obvious different chemical shift of the Si−H in THF-d8 and toluene-d8 support the coordination of **1a** with THF (for details, see

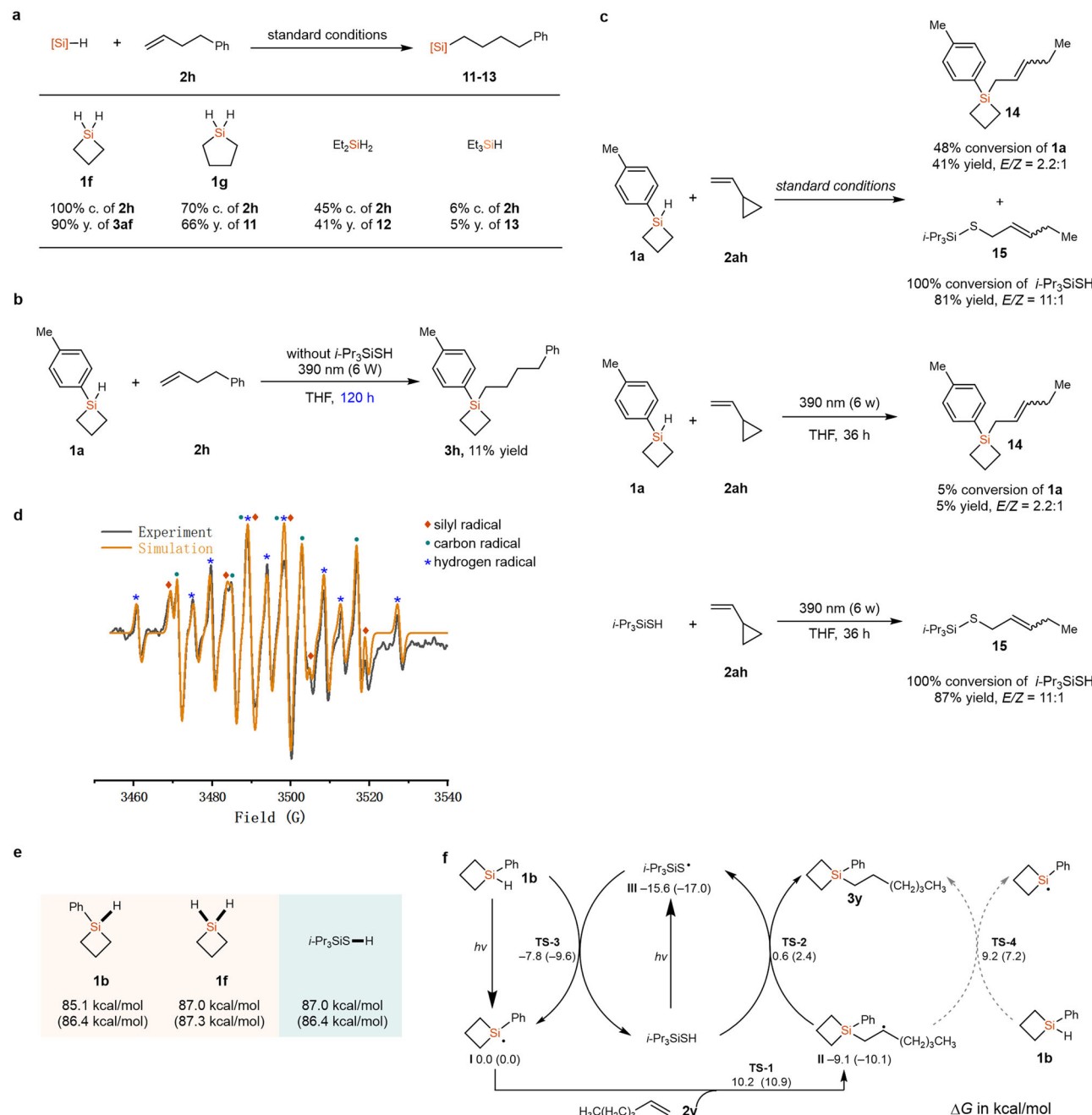

**Fig. 5 | Mechanistic study. a** Control experiments: non-strained cyclic and acyclic silanes under the standard conditions. "c.": conversion; "y.": yield. **b** Control experiment without *i*-Pr₃SiSH. **c** Radical clock experiment. **d** EPR study of **1a**. **e** The bond dissociation energies (Δ*H*) of Si/S−H bonds. **f** Proposed mechanism. The strained silyl radicals could be generated by direct irradiation and added to the alkene; the presence of the catalytic amount of *i*-Pr₃SiSH could accelerate the reaction through polarity-matched hydrogen atom transfer. Calculations were performed at the M06-2X/6-311G (d, p)/SMD (THF) level of theory with (data shown outside the parentheses) and without (data shown inside the parentheses) explicit THF. "**TS**": transition state.

Supplementary Figs. 5 and 6). However, we cannot rule out the stabilization of radical intermediate by polar solvent THF in facilitating the reaction. Subsequently, we employed density functional theory (DFT) to calculate the bond dissociation energies (BDEs) of the X−H bonds in compounds **1b**, **1f**, and the thiol. Figure 5e shows the BDEs with/without explicit THF. For compound **1b**, the Si−H BDE is 85.1 kcal/mol with explicit THF and 86.4 kcal/mol with implicit THF. For compound **1f**, the BDEs are 87.0 and 87.3 kcal/mol, respectively. For the S−H bond, the BDEs are 87.0 and 86.4 kcal/mol, respectively. Therefore, the presence of THF might affect the BDEs, notably for the Si−H bond in compound **1b** (1.3 kcal/mol difference), albeit this energy difference might be in

the error of the calculation method. Based on the above experimental results and literature reports[37–39,60], a plausible reaction mechanism is proposed for the hydrosilylation of unactivated alkenes with HSCBs, which is also supported by DFT calculations (Fig. 5f). HSCBs are excited by light to form silyl radicals **I**, which then selectively add to the less sterically hindered site of the unactivated alkenes via the transition state **TS-1** with an energy barrier of 10.2 kcal/mol, leading to carbon radicals **II**. The nucleophilic radicals **II** then undergo a polarity-matched hydrogen atom transfer (HAT) process via the transition state **TS-2** to afford the hydrosilylation product, along with the formation of thiyl radical **III**, with a total activation free energy of 9.7 kcal/mol. Thiyl

radical **III** could abstract the hydrogen atom from HSCBs to form silyl radicals **I** via transition state **TS-3**, with an activation free energy of 7.8 kcal/mol, while regenerating *i*-Pr$_3$SiSH. We also considered the HAT process of nucleophilic radical **II** with the HSCB through transition state **TS-4**, which could also produce the hydrosilylation product. However, the activation free energy for this step is 17.3 kcal/mol, which allows us to rule out this process in the reaction in the presence of a thiol catalyst. However, for the initiation of the reaction, we cannot rule out the possibility of first generating thiyl radical **III** from the direct irradiation of *i*-Pr$_3$SiSH followed by the HAT process to generate key intermediate **I**.

In conclusion, we have developed a visible-light-induced Lewis basic solvent-promoted metal-free silylation of unactivated alkenes with hydrosilacyclobutanes. The strained-ring silicon-centered radicals could be directly generated under visible-light irradiation without a redox-active photocatalyst. The thiol catalyst plays an important role in accelerating the reaction. A wide range of unactivated alkenes with diverse functional groups participate in this reaction smoothly, which significantly broadened the diversity and scope of silacyclobutanes. This unique metal-free single-electron process offers distinct advantages to the previous metal-based two-electron processes for the synthesis of SCBs. We believe the successful applications of hydrosilacyclobutanes in radical reactions and the ability to prepare various previously unreached highly functionalized SCBs will significantly promote the development of organosilicon chemistry.

## Methods

### General procedure for hydrosilylation of unactivated alkenes with hydrosilacyclobutanes (HCBs)

In an argon-filled glovebox, to a flame-dried screw-cap reaction tube equipped with a magnetic stir bar were added *i*-Pr$_3$SiSH (9.5 mg, 0.05 mmol, 10 mol%), hydrosilacyclobutanes (0.6 mmol, 1.2 equiv.), alkenes (0.5 mmol, 1 equiv.) and THF (2.0 mL) sequentially. The tube was sealed with a screw cap equipped with a septum, and removed from the glovebox. The reaction mixture was stirred at rt for 36 h under 6 W 390 nm LED lamps. The reaction mixture was concentrated under reduced pressure. The crude product was purified with column chromatography on silica gel to obtain the pure product.

## Data availability

All data needed to support the conclusions of this manuscript are included in the main text or supplementary information. Source Data are provided with this paper. All data are available from the corresponding author upon request. Source data are provided with this paper.

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

## Acknowledgements

We are grateful to the National Key R&D Program of China (2022YFA1506100, X.S.), the National Natural Science Foundation of China (22471201, 21901191, X.S.), the China Postdoctoral Science Foundation (2023TQ0252, 2023M742687, S.C.), the Postdoctoral Foundation of Hubei Province (211000032, S.C.) and the Postdoctoral Fellowship Program of CPSF (GZC20231960, S.C.) for financial support. The theoretical calculations were performed on the supercomputing system in the Supercomputing Center of Wuhan University.

## Author contributions

X.S. conceived the idea, guided the project and wrote the manuscript with revisions by all the other authors; S.C. developed the catalytic methods, performed the mechanistic studies and the synthetic applications. S.C. and M.G. prepared the substrates and studied the scope. X.H. performed the calculations. All the authors are involved in the discussion and analysis of the data.

## Competing interests

The authors declare no competing interests.
