## [Transparent Peer Review file · Nature Communications]

Photo-induced ring-maintaining hydrosilylation of unactivated alkenes with hydrosilacyclobutanes

Corresponding Author: Professor Xiao Shen

Version 0:

Reviewer comments:

Reviewer #1

(Remarks to the Author)

In this manuscript, Shen and co-workers report a visible-light-induced Lewis basic solvent promoted metal-free silylation of unactivated alkenes with hydrosilacyclobutanes for the synthesis of functionalized silacyclobutanes. Silacyclobutanes are a kind of interesting and versatile building blocks for the synthesis of a variety of organosilicon compounds. A long-standing hurdle for the investigation and application of silacyclobutanes is the lack of mild and efficient methods for the preparation of functionalized SCBs. The authors provide a remarkably simple and excellent solution to this problem. Most common functional groups including the base-sensitive acid, alcohol and ketones are well tolerated under the mild reaction conditions. In addition, unsymmetrical dialkyl SCBs can also be synthesized through consecutive hydrosilylation with DHSCB in one pot. The mechanism study reveals that the Lewis basic solvent could promote the generation of strained silyl radicals, which is distinct from conventional radical addition reactions between common hydrosilanes and alkenes. Overall, publication of this nice work in a prime journal such as Nature Communications is highly recommended after minor revisions as listed below.

1. Line 155, (3-methylbut-3-en-1-yl)benzene is not an internal alkene.
2. Are alkenes bearing EWG and styrene suitable for the reaction with the thiol catalyst instead of 4CzIPN and aceclidine (Fig 4d)?
3. Solvent effect seems significant in Table 1. The authors could add more common solvents with various coordination ability and give more comments on the reason of observed solvent effect. Perhaps comparison between the ¹H NMR of hydrosilacyclobutane in the presence and absence of Lewis basic solvent would be useful.
4. What is the result when common silanes such as non-strained cyclic and acyclic silanes are employed under the standard conditions? Comparison between the reactivity of hydrosilacyclobutanes and common hydrosilanes under the same conditions and discussion about the reason would be helpful for understanding the mechanism.

Jian Cao
Hangzhou Normal University

Reviewer #2

(Remarks to the Author)

Silacyclobutane is used in transformation reactions that use transition metal catalysts to activate Si-C bonds because it is more reactive than other cyclic silicon compounds due to the ring strain. Although the synthesis of silacyclobutane often involves a nucleophilic substitution reaction at the silicon centre, the high reactivity of the silacyclobutane ring to transition metals makes it difficult to use a transition metal catalyst for the synthesis of silacyclobutane. In this paper, Shen et al. report a third method for the synthesis of silylcyclobutane, a transformation reaction via a silyl radical using photocatalysis. Namely, the formal hydrosilylation of hydrosilanes with alkenes gives the corresponding hydrosilyl products. However, this basic reaction was already reported in 2023 by other researcher, Wu et al. (ref 37. Nat Chem, 2023, 666). Although this is a useful reaction, my impression is that this is a niche and not very original as it is an improvement on someone else's discovery. I don't think it's very suitable for a top journal like Nat Comm.

Reviewer #3

(Remarks to the Author)

NCOMMS-24-74684-T

Photo-induced ring-maintaining hydrosilylation of unactivated alkenes with hydrosilacyclobutanes

In this manuscript, the authors describe a general method to decorate silacyclobutanes with alkyl substituents based on an alkene hydrosilylation strategy that proceed under visible light.

In the first part, a concise overview of silacyclobutane chemistry is given. The importance of the silacyclobutane family is highlighted as well as the main synthetic routes to prepare such cyclic silicon species. As commented by the authors, these routes often involve chlorosilacyclobutanes with Grignard or lithium reagents. These organometallic species are indeed sensitive to moisture, which is not a real drawback nowadays ($n\text{BuLi}$ is used on the ton scale in industry), and they are not compatible with most functional groups which is, indeed, a real drawback. To circumvent their use, a Ni-catalyzed coupling was developed by Zhao and coworkers but this method does not permit the introduction of alkyl groups. Based on these considerations, the authors propose to transpose to silacyclobutanes a method developed for linear silanes: the radical hydrosilylation of alkenes.

In the second part called "Reaction development", the possibility of an hydrosilylation reaction between a Si-Aryl-Si-H-substituted hydrosilacyclobutane and an alkene possessing a remote acetyl group is assessed. The optimization of the reaction conditions is presented in the form of a table with entries corresponding to changes in photocatalyst type / solvent / light wavelength. Under optimized conditions, a selective hydrosilylation takes place with 390 nm light irradiation in presence of 10 mol% of $i\text{Pr}_3\text{SiSH}$ in THF solvent. The expected Si-Alkyl silacyclobutane could be isolated in 95 % yield. With the optimized conditions in hand, the next part of the manuscript is focused on the scope of the hydrosilylation reaction and its functional group tolerance. The reaction proved tolerant to a wide range of functional groups and it is noteworthy that free carbonyl as well as protic functions are tolerated under these conditions. A scope of 31 compounds could be prepared with globally good to excellent yields including in the case of structurally complex moieties such as L-menthol and DHEA derivatives.

To push forward the synthetic methodology, the use of Si,Si-dihydrosilacyclobutane in mono, first, and then double sequential hydrosilylation was then considered. This part is particularly relevant as the simplest congener of the silacyclobutane family are exploited for organic synthesis. In the case of the mono-hydrosilylation process, 12 different new compounds could be obtained in good to very good yields. Of note is the reaction with a terminal dialkene that leads selectively to the bis-silacyclobutane. The sequential double hydrosilylation afforded unsymmetrically substituted silacyclobutane and present the advantage to be done without isolation of the first hydrosilylated product.

The next part of the manuscript "Synthetic applications" is devoted to the post-transformation ("late-stage functionalization") of an unsymmetrically substituted silacyclobutane obtained with the new method described herein. The Si-Aryl-Si-Alkyl-substituted compound 3s, prepared on the gram scale in a proof of principle synthesis, is a direct precursor of 6-membered cyclic as well as acyclic silanes in transition-metal catalyzed reaction. This part shows that the classical ring-opening reactivity of silacyclobutane is maintained, and it connects the current methodology to previous work on silacyclobutane transformations. As a matter of fact, silicon with four different substituents can be obtained thanks to these combined methods.

The last part of the manuscript consists in a few control experiments carried out in order to unveil the mechanism of the hydrosilylation reaction. Stoichiometric experiments in presence of TEMPO suggest that both silyl and thyl radicals are formed under light irradiation, though the identity of the trapping products rely only on HR-MS analysis. The combination of these stoichiometric experiments with DFT-calculated BDE of X-H bonds leads the authors to propose a rather hypothetical mechanism.

The manuscript is well written, and the description of the different synthetic parts follows a logical order going from the mono-hydrosilylation reaction to the more complex sequential double-hydrosilylation to finish with a connection to previous methodologies in TM-catalyzed transformations. The hydrosilylation reaction described herein is very relevant in the context of silacyclobutane functionalization and more generally silicon chemistry and the possibility to use the simple Si,Si-dihydrosilacyclobutane as a starting material is really a plus both from a conceptual point of view and from a synthetic one. The power of the methodology is nicely illustrated with an important scope that should arouse interest of a broad range of chemists.

However, the Supporting Information is not fully convincing. Whereas clean NMR spectra for all the newly described compounds are provided, data is missing. First, no attribution is given in the NMR description of the compounds. Secondly, no ^{29}Si NMR data is provided which is problematic in a work focused on silicon chemistry. Finally, no melting points neither elemental analysis is provided. I understand that gathering elemental analysis for many compounds is a difficult task but it is an important purity criterium that should be considered for at least a few of the described compounds.

Additionally, the mechanistic part lacks clear evidence to support the proposed mechanism. First, the identification of the TEMPO trapping products based on HR-MS is not convincing. Characterization of the isolated trapping products would be beneficial in this context, and X-ray diffraction data would be the best. Additionally, the authors did not mention any attempts to observe radical species with EPR spectroscopy. Such experiments would also be highly beneficial. Finally, a complete DFT-computed mechanism would likely bring useful information on the overall mechanistic scheme.

As a consequence, I would not consider the present manuscript suitable for Nature Communications in the current form but it could be suitable if the authors can answer the above comments on the Supporting Informations and the Mechanistic part. Additionally, and with the aim to help the authors to improve the quality of the manuscript and the supporting information, I have listed under minor comments.

Manuscript

p.3, line 92 : "The reaction did not proceed without light irradiation, indicating the importance of light irradiation" This is redundant, please rephrase.

p.3, line 95 : The authors argue that this is the coordinative property of THF which is responsible for the reactivity. This is also simply the most polar solvent that has been tried in this study. Have the authors any clear proofs for such a coordination of THF? For example, is there any clear spectroscopic differences (for example in ^{29}Si NMR) between 1f in benzene or 1f in THF? The simple explanation could also be the stabilization of a polar intermediate by the more polar solvent.

p.3, line 100-103 : The authors did not mention if the chloro derivative of 1a coming from the reaction with dichloromethane was observed or isolated. Please comment.

p.5, line 144 : The preparation of 1f as a key compound is interesting and could be depicted in the manuscript. It is unclear if this compound was prepared or characterized before the current study. I could not find it somewhere else and there is no citation. Please cite relevant bibliography if the synthetic procedure has been adapted from a literature procedure. The yield of the synthesis (77 %) could be given in the main text for the interested reader. Also, if this compound was not characterized before, the ^{29}Si NMR of this compound should be given of course.

p.5, line 153 : Did the authors tried to prepare the spiro silane from 2g and 1f by playing with the experimental conditions (concentration for example)?

Supporting Informations

S35 : Whereas the authors argue in the main text that the coordinating role of THF is of prime importance, BDE calculations have been performed with a continuum model rather than explicit solvent molecules. Please comment.

Version 1:

Reviewer comments:

Reviewer #1

(Remarks to the Author)

The authors have exerted great effort to revise their manuscript in response to all comments of referees. The revisions and additional experiments carried out to strengthen this work are satisfying. No further major modifications seem necessary, at least not from this referee's perspective. I now recommend that the manuscript be published in Nature Communications.

Reviewer #3

(Remarks to the Author)

The authors have addressed the questions I had in the first round of evaluation and I consider now the present manuscript suitable for publication in Nature Communications in the current form with two comments that should be considered:

1) The authors argue that this is the coordinative property of THF which is responsible for the reactivity. To prove this hypothesis, the ^{29}Si NMR spectrum of the starting material 1f has been recorded in toluene and THF giving chemical shifts of -23.8 and -24.2 which can be considered as identical on a ^{29}Si NMR window. Additionally, the calculations on 1f with implicit and explicit solvent molecules give essentially the same results with energy difference in the error of the method (around 1 kcal.mol⁻¹ or even less). As a consequence, the results show no clear influence of the solvent. The calculations without solvent (PCM or explicit solvent molecules) has not been added for comparison and, as a consequence, it is hard to argue on the possible stabilization of radical intermediates by a polar solvent.

For sure is the fact that experimental data do not speak in favor of a coordinating THF molecule, that should be more clear in the manuscript.

2) It is not mentioned in the DFT calculated mechanism if ΔG or ΔH values are given.

Point-to-point response to the reviewers' comments

REVIEWER COMMENTS

Reviewer #1 (Remarks to the Author):

In this manuscript, Shen and co-workers report a visible-light-induced Lewis basic solvent promoted metal-free silylation of unactivated alkenes with hydrosilacyclobutanes for the synthesis of functionalized silacyclobutanes. Silacyclobutanes are a kind of interesting and versatile building blocks for the synthesis of a variety of organosilicon compounds. A long-standing hurdle for the investigation and application of silacyclobutanes is the lack of mild and efficient methods for the preparation of functionalized SCBs. The authors provide a remarkably simple and excellent solution to this problem. Most common functional groups including the base-sensitive acid, alcohol and ketones are well tolerated under the mild reaction conditions. In addition, unsymmetrical dialkyl SCBs can also be synthesized through consecutive hydrosilylation with DHSCB in one pot. The mechanism study reveals that the Lewis basic solvent could promote the generation of strained silyl radicals, which is distinct from conventional radical addition reactions between common hydrosilanes and alkenes.

Overall, publication of this nice work in a prime journal such as Nature Communications is highly recommended after minor revisions as listed below.

Our response: Thanks for the positive recommendation.

1. Line 155, (3-methylbut-3-en-1-yl)benzene is not an internal alkene.

Our response: We are sorry for this mistake. We have modified “The reactions with internal alkenes such as (*Z*)-cyclooctene and (3-methylbut-3-en-1-yl)benzene worked well, giving corresponding hydrosilylation products **3ap** and **3aq** in 50% and 67% yield respectively.” into “The reactions with internal alkene such as (*Z*)-cyclooctene and 1,1-disubstituted alkene such as (3-methylbut-3-en-1-yl)benzene worked well, giving corresponding hydrosilylation products **3ap** and **3aq** in 50% and 67% yield respectively.”

2. Are alkenes bearing EWG and styrene suitable for the reaction with the thiol catalyst instead of 4CzIPN and aceclidine (Fig 4d)?

Our response: Thanks for the comments. Unfortunately, alkenes bearing EWG and styrene are not suitable for the reaction with the thiol catalyst instead of 4CzIPN and aceclidine. The radicals generated via the addition of silyl radicals to the alkenes bearing EWG are electrophilic radicals, which are polarity mismatched with thiol catalyst in the HAT process. These radicals are prone to be reduced to carbanions under the photocatalytic conditions in the presence of 4CzIPN and aceclidine. As for styrene, the benzyl radical generated via the silyl radical addition is a stabilized radical which is probably difficult to undergo HAT with a thiol.

3. Solvent effect seems significant in Table 1. The authors could add more common

solvents with various coordination ability and give more comments on the reason of observed solvent effect. Perhaps comparison between the ^1H NMR of hydrosilacyclobutane in the presence and absence of Lewis basic solvent would be useful.

Our response: Thanks for the kind suggestions. We have tested DMSO, DMF, CH_3CN and acetone as the solvent. However, these solvents are not as good as THF. We have added these data in the revised supporting information.

Entry	HAT catalyst	Solvent	conv. of 1a	Yield of 3a
1	i -Pr ₃ SiSH	DMSO	2%	trace
2	i -Pr ₃ SiSH	DMF	5%	3%
3	i -Pr ₃ SiSH	CH ₃ CN	8%	4%
4	i -Pr ₃ SiSH	Acetone	45%	38%

[a] Reaction conditions: **1a** (0.1 mmol), **2a** (0.1 mmol), HAT catalyst (10 mol%), light (6 W), solvent (1 mL), 12 h.

Following the reviewer's suggestion, we have compared the ^1H NMR of **1a** in THF- d_8 and Toluene- d_8 (with tetramethylsilane as an internal standard). We found that when the THF was used, the chemical shift of the Si-H decreased, indicating that THF may coordinate with hydrosilacyclobutane. We have added this data in the revised manuscript and supplementary information.

4. What is the result when common silanes such as non-strained cyclic and acyclic silanes are employed under the standard conditions? Comparison between the reactivity of hydrosilacyclobutanes and common hydrosilanes under the same conditions and discussion about the reason would be helpful for understanding the mechanism.

Our response: Thanks for the kind suggestions. Under the standard conditions, non-strained silacyclopentane (**1g**) has been tried, both the conversion of **2h** and the yield of product decreased, compared with the reaction of **1f**. Acyclic silanes such as Et₂SiH₂ has also been tried, both the conversion of **2h** and the yield of product decreased, compared with the reaction of **1f**. The reactivity of Et₃SiH is much lower, generating hydrosilylation product in only 5% yield. These results indicate that the existence of ring leads to higher reactivity of the silicon-hydrogen bond of HSCB.

Jian Cao
Hangzhou Normal University

Reviewer #2 (Remarks to the Author):

Silacyclobutane is used in transformation reactions that use transition metal catalysts to activate Si-C bonds because it is more reactive than other cyclic silicon compounds due to the ring strain. Although the synthesis of silacyclobutane often involves a nucleophilic substitution reaction at the silicon centre, the high reactivity of the silacyclobutane ring to transition metals makes it difficult to use a transition metal catalyst for the synthesis of silacyclobutane. In this paper, Shen et al. report a third method for the synthesis of silylcyclobutane, a transformation reaction via a silyl radical using photocatalysis. Namely, the formal hydrosilylation of hydrosilanes with alkenes gives the corresponding hydrosilyl products. However, this basic reaction was already reported in 2023 by other resercher, Wu et al. (ref 37. Nat Chem, 2023, 666). Although this is a useful reaction, my impression is that this is a niche and not very original as it is an improvement on someone else's discovery. I don't think it's very suitable for a top journal like Nat Comm.

Our response: Thanks for the comments. We regret that we did not succeed in convincing Reviewer 2 of the novelty of our findings. However, previous radical hydrosilylation require redox-active photocatalyst to induce the generation of silyl radicals and cooperative catalysis with photocatalyst and HAT catalyst were used to

achieve the reported transformation. Wu's seminal work was limited to linear hydrosilanes, and the radical hydrosilylation with hydrosilacyclobutanes are more challenging because of the labile C-Si bonds of the strained four-membered ring compounds. In our work, we disclose that the strained-ring silicon-centered radicals could be directly generated under visible-light-irradiation without a redox-active photocatalyst, and the thiol catalyst plays an important role in accelerating the reaction. The reaction conditions are simpler than the previous work. Notably, under the Wu's conditions (ref 37. *Nat Chem*, **15**, 666 (2023); Table 1, entry 1 of our manuscript), our reaction couldn't proceed well because of the significant decomposition of MHSCB **1a**.

In addition, we also provide insight of the radical hydrosilylation mechanism. The combined experimental and DFT calculation supported that the coordinating THF could promote the generation of silyl radicals from the hydrosilacyclobutanes. Notably, we observed the formation of silyl radical through EPR studies. Moreover, the detailed DFT calculation also bring useful information on the overall mechanistic scheme. Last but not least, our findings provide an unprecedented access to the silacyclobutanes that are difficult to prepare otherwise. We believe the successful applications of hydrosilacyclobutanes in radical reactions and the ability to prepare various previously unreached highly functionalized SCBs will significantly promote the development of organosilicon chemistry.

Reviewer #3 (Remarks to the Author):

NCOMMS-24-74684-T

Photo-induced ring-maintaining hydrosilylation of unactivated alkenes with hydrosilacyclobutanes

In this manuscript, the authors describe a general method to decorate silacyclobutanes with alkyl substituents based on an alkene hydrosilylation strategy that proceed under visible light.

In the first part, a concise overview of silacyclobutane chemistry is given. The importance of the silacyclobutane family is highlighted as well as the main synthetic routes to prepare such cyclic silicon species. As commented by the authors, these routes often involve chlorosilacyclobutanes with Grignard or lithium reagents. These organometallic species are indeed sensitive to moisture, which is not a real drawback nowadays (nBuLi is used on the ton scale in industry), and they are not compatible with most functional groups which is, indeed, a real drawback. (**Our response:** Thanks for the comments and kind suggestions, we have deleted "moisture sensitive" in the revised manuscript) To circumvent their use, a Ni-catalyzed coupling was developed by Zhao and coworkers but this method does not permit the introduction of alkyl groups. Based on these considerations, the authors propose to transpose to silacyclobutanes a method developed for linear silanes: the radical hydrosilylation of alkenes.

In the second part called "Reaction development", the possibility of an hydrosilylation reaction between a Si-Aryl-Si-H-substituted hydrosilacyclobutane and an alkene

possessing a remote acetyl group is assessed. The optimization of the reaction conditions is presented in the form of a table with entries corresponding to changes in photocatalyst type / solvent / light wavelength. Under optimized conditions, a selective hydrosilylation takes place with 390 nm light irradiation in presence of 10 mol% of *i*Pr₃SiSH in THF solvent. The expected Si-Alkyl silacyclobutane could be isolated in 95 % yield.

With the optimized conditions in hand, the next part of the manuscript is focused on the scope of the hydrosilylation reaction and its functional group tolerance. The reaction proved tolerant to a wide range of functional groups and it is noteworthy that free carbonyl as well as protic functions are tolerated under these conditions. A scope of 31 compounds could be prepared with globally good to excellent yields including in the case of structurally complex moieties such as L-menthol and DHEA derivatives.

To push forward the synthetic methodology, the use of Si,Si-dihydrosilacyclobutane in mono, first, and then double sequential hydrosilylation was then considered. This part is particularly relevant as the simplest congener of the silacyclobutane family are exploited for organic synthesis. In the case of the mono-hydrosilylation process, 12 different new compounds could be obtained in good to very good yields. Of note is the reaction with a terminal dialkene that leads selectively to the bis-silacyclobutane. The sequential double hydrosilylation afforded unsymmetrically substituted silacyclobutane and present the advantage to be done without isolation of the first hydrosilylated product.

The next part of the manuscript “Synthetic applications” is devoted to the post-transformation (“late-stage functionalization”) of an unsymmetrically substituted silacyclobutane obtained with the new method described herein. The Si-Aryl-Si-Alkyl-substituted compound 3s, prepared on the gram scale in a proof of principle synthesis, is a direct precursor of 6-membered cyclic as well as acyclic silanes in transition-metal catalyzed reaction. This part shows that the classical ring-opening reactivity of silacyclobutane is maintained, and it connects the current methodology to previous work on silacyclobutane transformations. As a matter of fact, silicon with four different substituents can be obtained thanks to these combined methods.

The last part of the manuscript consists in a few control experiments carried out in order to unveil the mechanism of the hydrosilylation reaction. Stoichiometric experiments in presence of TEMPO suggest that both silyl and thyl radicals are formed under light irradiation, though the identity of the trapping products rely only on HR-MS analysis. The combination of these stoichiometric experiments with DFT-calculated BDE of X–H bonds leads the authors to propose a rather hypothetical mechanism.

The manuscript is well written, and the description of the different synthetic parts follows a logical order going from the mono-hydrosilylation reaction to the more complex sequential double-hydrosilylation to finish with a connection to previous methodologies in TM-catalyzed transformations. The hydrosilylation reaction described herein is very relevant in the context of silacyclobutane functionalization and more generally silicon chemistry and the possibility to use the simple Si,Si-dihydrosilacyclobutane as a starting material is really a plus both from a conceptual

point of view and from a synthetic one. The power of the methodology is nicely illustrated with an important scope that should arouse interest of a broad range of chemists.

However, the Supporting Information is not fully convincing. Whereas clean NMR spectra for all the newly described compounds are provided, data is missing. First, no attribution is given in the NMR description of the compounds. Secondly, no ^{29}Si NMR data is provided which is problematic in a work focused on silicon chemistry. Finally, no melting points neither elemental analysis is provided. I understand that gathering elemental analysis for many compounds is a difficult task but it is an important purity criterium that should be considered for at least a few of the described compounds.

Our response: Thanks for the comments and kind suggestions. The attribution of characteristic ^1H and ^{13}C signals have been added in the NMR description. The ^{29}Si NMR data of all new compounds have been added. We have checked all the new compounds, only **3x** is a solid, and melting point of **3x** has been added. The elemental analysis of **3g** has been added. Since the NMR spectra can provide purity information of the product, we decided not to test the elemental analysis of other products.

Additionally, the mechanistic part lacks clear evidence to support the proposed mechanism. First, the identification of the TEMPO trapping products based on HR-MS is not convincing. Characterization of the isolated trapping products would be beneficial in this context, and X-ray diffraction data would be the best. Additionally, the authors did not mention any attempts to observe radical species with EPR spectroscopy. Such experiments would also be highly beneficial.

Our response: Thanks for the comments and kind suggestions. Since we can not get enough TEMPO trapping products to perform X-ray diffraction data analysis, we have deleted this experimental data in the revised manuscript. Instead, we used compound **2ah** to conduct a radical clock experiments. When the reaction was performed under the standard conditions, compounds **14** was isolated in 41% yield, and compound **15** was isolated in 81% yield, supporting the generation of a silyl radical and a thiyl radical in the reaction. Without *i*-PrSiSH, compound **14** was also isolated in 5% yield. In addition, the reaction of *i*-PrSiSH and **2ah**, in the absence of **1a**, afforded **15** in 85% yield.

EPR spectroscopy has been conducted with **1a**, *i*-PrSiSH and reaction system separately (with DMPO as radical spin trapping reagent):

A. EPR studies of 1a: A mixture signal of **radical 1** ($g = 2.0076$, $A_N = 14.30$ G, $A_H = 20.60$ G), **carbon radical** ($g = 2.0076$, $A_N = 13.90$ G, $A_H = 18.0$ G) and **hydrogenated DMPO** ($g = 2.0076$, $A_N = 14.50$ G, $A_{H1} = 18.80$ G, $A_{H2} = 18.80$ G) were identified. The simulation data for **carbon** and **hydrogen radicals** are consistent with reference (*Free Radical Bio. Med.* **1987**, *3*, 259–303. Page 264, line 40; Page 261). Based on radical clock experiments and **EPR studies of *i*-Pr₃SiSH**, We speculated that **radical 1** is a **silyl radical**. **Carbon radical** may be formed from the HAT process of **silyl radical** or **hydrogen radical** with THF.

B. EPR studies of *i*-Pr₃SiSH: A mixture signal of **thiyl radical** ($g = 2.0083$, $A_N = 13.20$ G, $A_H = 11.60$ G), **carbon radical** ($g = 2.0078$, $A_N = 13.90$ G, $A_H = 18.0$ G) were identified. The simulation data for **thiyl radical** and **carbon radical** are consistent with reference (*Free Radical Bio. Med.* **1987**, *3*, 259–303. Page 272, line 21–23; Page 264, line 40). **Carbon radical** may be formed from the HAT process of **thiyl radical** with THF.

C. EPR studies of the reaction system: A mixture signal of **radical 1** ($g = 2.0079$, $A_N = 14.30$ G, $A_H = 20.60$ G), **thiyl radicals** ($g = 2.0083$, $A_N = 13.20$ G, $A_H = 11.60$ G), and **carbon radicals** ($g = 2.0079$, $A_N = 13.90$ G, $A_H = 18.0$ G) was identified. The experimental result provided strong evidence for the radical process of the reaction.

Finally, a complete DFT-computed mechanism would likely bring useful information on the overall mechanistic scheme.

Our response: Thanks for the comments and kind suggestions. Based on the reviewer's

suggestion, we performed DFT calculation on the possible mechanism. We reasoned that THF might coordinate to the silacyclobutane and promote the homolysis of the Si-H bond. We performed an analysis using explicit THF molecules. As shown in Fig. 5f, a plausible reaction mechanism is proposed for the hydrosilylation of unactivated alkenes with HSCBs. HSCBs are excited by light to form silyl radicals **I**, which then selectively add to the less sterically hindered site of the unactivated alkenes via the transition state **TS-1** with an energy barrier of 10.2 kcal/mol, leading to produce carbon radicals **II**. The nucleophilic radicals **II** then undergo polarity-matched hydrogen atom transfer (HAT) process via the transition state **TS-2** to afford the hydrosilylation product, along with the formation of thiyl radical **III**, with a total activation free energy of 9.7 kcal/mol. Thiyl radical **III** could abstract the hydrogen atom from HSCBs to form silyl radicals **I** via transition state **TS-3**, with an activation free energy of 7.8 kcal/mol, while regenerating *i*-Pr₃SiSH. In addition, we also considered the nucleophilic radical **II** undergoing a HAT process with HSCBs through the transition state **TS-4**, resulting in the formation of a hydrosilylation product and a silyl radicals **I**. The activation free energy for this step is 17.3 kcal/mol. The DFT calculation data is consistent with the experimental fact that the reaction could proceed slowly in the absence of the thiol catalyst (Fig. 5b), but the thiol catalyst could significantly accelerate the reaction. However, for the initiation of the reaction, we cannot rule out the possibility of firstly generating thiyl radical **III** from the direct irradiation of *i*-Pr₃SiSH followed by HAT process to generate key intermediate **I**.

Fig. 5. Mechanistic study. f) Proposed mechanism. Calculations were performed at the M06-2X/6-311G (d, p)/SMD (THF) level of theory with (outside the parentheses) and without (inside the parentheses) explicit THF. The values are shown in kcal/mol.

As a consequence, I would not consider the present manuscript suitable for Nature Communications in the current form but it could be suitable if the authors can answer the above comments on the Supporting Informations and the Mechanistic part. Additionally, and with the aim to help the authors to improve the quality of the manuscript and the supporting information, I have listed under minor comments.

 Manuscript

p.3, line 92 : "The reaction did not proceed without light irradiation, indicating the importance of light irradiation" This is redundant, please rephrase.

Our response: Thanks for the kind suggestions. We have modified “The reaction did not proceed without light irradiation, indicating the importance of light irradiation” into “The reaction did not proceed without light irradiation.”

p.3, line 95 : The authors argue that this is the coordinative property of THF which is responsible for the reactivity. This is also simply the most polar solvent that has been tried in this study. Have the authors any clear proofs for such a coordination of THF? For example, is there any clear spectroscopic differences (for example in ^{29}Si NMR) between **1f** in benzene or **1f** in THF? The simple explanation could also be the stabilization of a polar intermediate by the more polar solvent.

Our response: Thanks for the kind suggestions. We have compared the ^{29}Si NMR of **1f** in THF-d₈ and Toluene-d₈ (with tetramethylsilane as internal standard), found that there was a slight spectroscopic differences. We have also compared the ^1H NMR of **1a** in THF-d₈, and Toluene-d₈ (with tetramethylsilane as an internal standard). We found that when the Lewis basicity of the solvent increased, the chemical shift of the Si-H decreased. These results indicate that hydrosilacyclobutane probably coordinate with THF. We can not rule out the stabilization of radical intermediate by polar solvent THF. We are grateful to the reviewer’s kind suggestion and we have added this possibility in the revised manuscript.

p.3, line 100-103 : The authors did not mention if the chloro derivative of **1a** coming from the reaction with dichloromethane was observed or isolated. Please comment.

Our response: Thanks for the kind suggestions. The chloro derivative of **1a** coming from the reaction with dichloromethane was observed by ^{29}Si NMR, the product was quenched by MeMgBr and compound **1a-1** was also isolated in 63% yield. We have added this data in the revised manuscript and supplementary information.

p.5, line 144 : The preparation of **1f** as a key compound is interesting and could be depicted in the manuscript. It is unclear if this compound was prepared or characterized before the current study. I could not find it somewhere else and there is no citation. Please cite relevant bibliography if the synthetic procedure has been adapted from a literature procedure. The yield of the synthesis (77 %) could be given in the main text for the interested reader. Also, if this compound was not characterized before, the ^{29}Si NMR of this compound should be given of course.

Our response: Thanks for the kind suggestions. We found no characterization data of compound **1f** from the literature, and we developed the synthesis method of **1f** by ourselves. The 77% yield of **1f** has been added in the main text. The ^{29}Si NMR of **1f** has been added in SI.

p.5, line 153: Did the authors tried to prepare the spirosilane from **2g** and **1f** by playing with the experimental conditions (concentration for example)?

Our response: Thanks for the kind suggestions. We wonder whether the reviewer was asking to prepare a spirosilane from diene **2o** (not **2g**) and **1f**. Unfortunately, a complex mixture of monohydrosilylation product, bishydrosilylation product, possible spiroilane and oligopolymerization product were detected. The polarity of the monohydrosilylation, bishydrosilylation and possible spiroilane are similarly small, thus we failed get pure product.

Supporting Informations

S35 : Whereas the authors argue in the main text that the coordinating role of THF is of prime importance, BDE calculations have been performed with a continuum model rather than explicit solvent molecules. Please comment.

Our response: Thanks for the comments and kind suggestions. Based on the reviewer's suggestion, we performed an analysis using explicit THF molecules. The bond dissociation energies (BDEs) with/without explicit THF are shown in Fig. 5e. Specifically, for compound **1b**, the Si-H BDE is 85.1 kcal/mol with explicit THF and 86.4 kcal/mol with implicit THF. For compound **1f**, the BDEs are 87.0 kcal/mol with explicit THF and 87.3 kcal/mol with implicit THF. For the S-H bond, the BDEs are 87.0 kcal/mol with explicit THF and 86.4 kcal/mol with implicit THF. These data indicate

that the presence of THF could affect the BDEs, particularly for the Si-H bond in compound **1b**, where the difference is 1.3 kcal/mol.

Fig. 5. Mechanistic study. e) BDEs of Si/S-H bonds. Calculations were performed at the M06-2X/6-311G (d, p)/SMD (THF) level of theory with (outside the parentheses) and without (inside the parentheses) explicit THF.

Point to point responses (NCOMMS-24-74684A)

REVIEWERS' COMMENTS

Reviewer #1 (Remarks to the Author):

The authors have exerted great effort to revise their manuscript in response to all comments of referees. The revisions and additional experiments carried out to strengthen this work are satisfying. No further major modifications seem necessary, at least not from this referee's perspective. I now recommend that the manuscript be published in Nature Communications.

Our response: Thanks for the positive recommendation.

Reviewer #3 (Remarks to the Author):

The authors have addressed the questions I had in the first round of evaluation and I consider now the present manuscript suitable for publication in Nature Communications in the current form with two comments that should be considered:

1) The authors argue that this is the coordinative property of THF which is responsible for the reactivity. To prove this hypothesis, the ^{29}Si NMR spectrum of the starting material 1f has been recorded in toluene and THF giving chemical shifts of -23.8 and -24.2 which can be considered as identical on a ^{29}Si NMR window. Additionally, the calculations on 1f with implicit and explicit solvent molecules give essentially the same results with energy difference in the error of the method (around 1 kcal.mol⁻¹ or even less). As a consequence, the results show no clear influence of the solvent. The calculations without solvent (PCM or explicit solvent molecules) has not been added for comparison and, as a consequence, it is hard to argue on the possible stabilization of radical intermediates by a polar solvent.

For sure is the fact that experimental data do not speak in favor of a coordinating THF molecule, that should be more clear in the manuscript.

Our response: Thanks for the comments. We have also compared the ^1H NMR of **1a** in THF-d₈, and Toluene-d₈ (with tetramethylsilane as an internal standard). We found that when the Lewis basicity of the solvent increased, the chemical shift of the Si-H decreased. These results indicate that hydrosilacyclobutane may have a weak coordination with THF, which is consistent to the calculated small calculated energy difference (1.3 kcal.mol⁻¹ or less).

We have used the radical stabilization energy (RSE, *J. Phys. Chem. A* **2001**, *105*, 6750; *J. Phys. Chem. A* **2007**, *111*, 13638) to compare the stability of the silyl radical intermediates in the gas phase and in the THF solvent. The results are as follows: in the gas phase, the RSE is -17.8 kcal/mol; in the THF solvent, the RSE is -18.9 kcal/mol. These results indicate that the silyl radical is more stable in the THF solvent, with an RSE value that is 1.1 kcal/mol lower than that in the gas phase. Therefore, we can not rule out the stabilization of radical intermediate by polar solvent THF in facilitating the reaction.

2) It is not mention in the DFT calculated mechanism if deltaG or deltaH values are given.

Our response: We are thankful for reviewer's comments, which help us improve the quality of manuscript. In our study, the bond dissociation energies (BDEs) are calculated as the enthalpy change (ΔH) at 298.15 K, representing the energy required to break a bond in the absence of entropy contributions (Fig. 5e). In addition, we use Gibbs free energy (ΔG) to assess the feasibility of the reaction (Fig. 5f). We have updated the manuscript to clearly state this distinction and ensure that our readers understand the context in which each value is used.